materials science

biomaterials, Ti-based alloy, alloying, trace precipitation, corrosion resistance, density functional theory calculation

**Author for correspondence:**
Zong-Yan Zhao
e-mail: zzy@kust.edu.cn

This article has been edited by the Royal Society of Chemistry, including the commissioning, peer review process and editorial aspects up to the point of acceptance.

# Effect of Ag alloying and trace precipitation on corrosion resistance of Ti-Ta-Ag ternary alloy

Jun-Min Zhang, Zong-Yan Zhao, Qing-Hua Chen, Xing-Hu Chen and Yin-He Li

Faculty of Materials Science and Engineering, Kunming University of Science and Technology, Kunming, Yunnan 650093, People's Republic of China

Z-YZ, 0000-0001-8953-3150

This work systematically analysed the electrochemical and corrosion behaviour of Ti-Ta-Ag ternary alloy samples in Hank's solution. For the samples with 1.5% and 3% Ag content, the sintering temperature increased from 750 to 950°C, and the corresponding corrosion resistance increased by 100 times due to the increased alloying of Ag; meanwhile for the sample with 4.5% Ag content, the sintering temperature increased from 750 to 950°C, and the corresponding corrosion resistance decreased by six times due to the increased precipitation of Ag. These tests prove that the Ag alloying is beneficial to the enhancement of the corrosion resistance of Ti-Ta-Ag ternary alloy, but the Ag trace precipitation has the opposite effect. A series of electrochemical characterizations and density functional theory calculations explain the mechanism of the above phenomenon. Ag alloying can promote the formation of uniform, complete, dense, stable and thick passivation layer on the surface of Ti-Ta-Ag ternary alloy, which makes Ti-Ta-Ag ternary alloy uniformly corroded without pitting. In addition, Ag alloying can effectively reduce the contact resistance of the solid–liquid interface. However, the trace precipitation of Ag plays the opposite role to the above effect.

## 1. Introduction

Since it was used as a dental implant by Branemark *et al.* in the 1960s, titanium (Ti) and Ti-based alloys have been widely used in the field of biology as surgical implant materials [1,2]. The most outstanding advantages of Ti and Ti-based alloys are small specific gravity, high specific strength, low elastic modulus, excellent corrosion resistance and good biocompatibility, which are the essential reasons why they become the first choice for

biomedical metal materials [3,4]. A large numbers of research groups and scholars have conducted in-depth research on the corrosion of medical Ti-based alloys and their influencing factors, and achieved a lot of meaningful results. It has been found that pure Ti metal has excellent corrosion resistance because it can form a compact and stable oxide film. But its poor mechanical properties lead to poor wear resistance in practical application [5]. In order to enhance the mechanical properties and corrosion resistance of pure Ti metal, a series of Ti-based alloys have been developed and used. For example, Ti-Ni [6–8], Ti-Ta [9], Ti-15Mo [10], Ti-Nb-Sn [11], Ti-12Mo-5Ta [12], Ti-29Nb-13Ta-4.6Zr [10] and so on, which have shown good corrosion resistance. However, the alloying of Ti with some metal components may bring adverse effects, for example Ti-6Al-4 V has been proved to have potential toxic and side-effects [13].

Among the developed Ti-based alloys, the performance of Ti-Ta alloy as a biomedical metal material attracts special attention, because of its excellent mechanical properties and corrosion resistance. Zhou *et al*. [14] and Mareci *et al*. [15] independently found that Ti-Ta alloy with 10–70 wt% of Ta content has excellent corrosion resistance in simulated human body solution. By analysing the anodic polarization curve of Ti-Ta alloy in 5% hydrochloric acid solution and the surface structure after corrosion, Ti-Ta alloy has better corrosion resistance than pure Ti. With the increase of Ta content, the corrosion current of Ti-Ta alloy decreases and the breakdown potential increases. At the same time, Ti-Ta alloy with 25 wt% of Ta content has the lowest elastic modulus and the highest strength/elastic modulus ratio [14]. In the previous work, we systematically analysed the structure and properties of Ti-Ta solid solution by density functional theory (DFT) calculations and found that the elastic modulus and shear modulus of Ti-Ta alloy have different performance in different solid solution range, which is determined by the stable structure, bonding type and electronic contribution [16].

In order to further improve the antibacterial properties and biocompatibility of Ti-Ta alloy, the preparation of Ti-Ta-M ternary alloys by adding precious metal elements has become a research hotspot. As a broad-spectrum germicidal material, Ag has excellent antibacterial properties [17,18]. Although the antibacterial mechanism of Ag has not been fully studied up to now, some studies have shown that Ag nanoparticles have excellent antibacterial effects on four common bacteria in orthopaedic surgery, such as Staphylococcus Aureus—the most common pathogenic bacteria in orthopaedic surgery—Staphylococcus Epidermidis, Escherichia Coli and Klebsiella Pneumoniae [17,18]. Hou *et al*. have carried out studies on biological Ti-based alloys containing Ag [19–22]. In the Ti-Ni-Ag alloy prepared by the arc melting method, Ag precipitates in irregular micron-size particles. The shape memory effect of Ti-Ni-Ag alloy is almost not affected by the addition of trace amounts of Ag, and the antimicrobial activity of the Ti-Ni-Ag alloy is obviously improved [19]. Moreover, the addition of trace Ag can effectively improve the corrosion resistance of pure Ti in the artificial saliva solution [20]. The porous Ti-3Ag alloy was prepared by ball milling mixed Ti and Ag powders, in which the refining of powder can lead to the increase of surface energy, thus reducing the Ag particles precipitated on the surface [21,22]. In addition, Ti-Ag alloy showed higher corrosion potential and greater impedance than CP-Ti alloys; [23,24] the Ti$_2$Ag phase exhibited strong dispersion strengthening performance in Ti-Ag alloy, in which Ag alloying can improve the corrosion resistance better than Ag precipitating; [25] Ag alloying in Ti-based alloys can reduce the Young's modulus and increase the antibacterial activity [26]. These cases almost without exception have proven that Ag alloying can significantly improve the antibacterial properties and biocompatibility of Ti-Ta-Ag alloys, which makes Ti-Ta-Ag alloys one of the new biomedical Ti-based alloys with good comprehensive performance.

In our previous work, we found Ti-Ta-Ag can form a stable ternary alloy system by spark plasma sintering (SPS), and Ag will be precipitated at the grain boundary with the increase of Ag content and sintering temperature. The internal mechanism of this phenomenon is that the 4d$^{10}$ electronic states of Ag have changed from obvious local electronic states to mixed local and non-local electronic states [27]. However, as described above, the effects of Ag precipitation in Ti-Ta-Ag alloy on corrosion resistance and its internal mechanism are not clearly understood, which is vital for its biomedical applications. Therefore, on the basis of previous research, in the present work we focused on the analysis of the influence of Ag alloying and precipitating on the corrosion resistance of Ti-Ta-Ag ternary alloy, and systematically analysed the internal influence mechanism by DFT calculations.

## 2. Method details

The samples used in this work are those we prepared in the previous work [27]. Their preparation process is summarized as follows. The raw materials used for the preparation of samples were Ti, Ta and Ag metal powders with purity greater than 99.5%, and their particle sizes were less than or equal

to 55, less than or equal to 30, 7–10 µm, respectively. First, Ti, Ta and Ag metal powders were respectively weighed according to the mass fraction of $(75 - x)\%$, 25% and $x\%$ ($x = 0$, 1.5, 3 and 4.5). The weighed metal powders were mixed with $ZrO_2$ as a ball milling medium (the ratio of ball to material was maintained at 3.5 : 1), and then put into the vacuum ball milling tank. The vacuum ball mill tank was sealed and vacuumed to approximately 10 Pa, and then filled with argon as the protection gas. Then the mixed metal powders were ball milled at 150 RPM for 20 h. Finally, the fully mixed ball milled metal powders were loaded into the graphite mould (Ø20 × 50 mm) and sintered in the SPS furnace. Sintering process parameters were set as follows: the temperature was raised to the desired temperatures (750, 850 and 950°C) at a heating rate of 90°C min$^{-1}$, the axial pressure of 30 MPa was continuously applied in the sintering process, the vacuum degree of the system was 2–8 Pa and the sintering time was 5 min. After the pulse current was turned off, the samples were quickly cooled to room temperature in the furnace and taken out for subsequent testing.

## 2.1. Electrochemical behaviour

The fabrication process of the working electrode was as follows: (i) the surface of the alloy sample sintered by SPS is polished layer by layer with metallographic sandpaper, and then mechanically polished to make the surface of the sample bright; (ii) copper wire (2.5 mm$^2$) was welded on the one side of the alloy sample by the conductive adhesive tape, to ensure that the contact resistance between them was 0 Ω; (iii) the prepared electrode was inlaid with epoxy resin, that is, the alloy surface without copper wire connection is exposed, and the rest is completely sealed in the epoxy resin to ensure that there is no gap between the alloy sample surface and the sealed epoxy resin; (iv) the fabricated working electrode was mechanically polished again to remove the residual epoxy resin on the surface, and then it was cleaned with acetone ultrasonic and thoroughly cleaned with alcohol; finally, it was naturally air-dried and put into a sealed bag to be tested. In this work, the working electrodes were wafers with a diameter of about 20 mm.

A multifunctional electrochemical workstation (ModuLab XM ECS, Solartron analytical, UK) was used for corrosion resistance testing in the present work. The three-electrode comparison method was used in the corrosion experiment: saturated calomel electrode (SCE) was selected as the reference electrode, platinum electrode as the auxiliary electrode and Ti-Ta-Ag alloy sample as the working electrode. After the above three electrodes were installed in a fixed position, they were put into a quartz electrolytic cell filled with artificial body fluids. The artificial simulated body fluid used in experimental tests was Hank's solution. Its composition was as follows: NaCl (8.0 g), KCl (0.4 g), NaHCO$_3$ (0.35 g), Na$_2$HPO$_4$ · 12H$_2$O (0.12 g), KH$_2$PO$_4$ (0.06 g) and glucose (1.0 g). The above ingredients were added into deionized water (total volume: 1 l) and prepared in Hank's solution with a concentration of 1 mol l$^{-1}$. The working electrode, auxiliary electrode and reference electrode were soaked in Hank's solution for 30 min before testing. The conditions of the electrochemical test were set as follows: the test time of open-circuit voltage was 6000 s; the potential scanning range of the polarization curve was -2–2 V and the scanning speed was 5 mV s$^{-1}$; the amplitude of electrochemical impedance spectroscopy was 10 mV and the frequency range was $10^{-1}$–$10^5$ Hz.

## 2.2. Density functional theory calculation

Based on the DFT calculation of (Ti$_{1-x}$Ag$_x$)$_3$Ta solid solution and the findings of experimental research in our previous work [27], we take (Ti$_{1-x}$Ag$_x$)$_3$Ta ($x = 0$ and 0.083) as the research object in the present work. The surface model was cleaved along with the (001) crystal plane and constructed by 2 × 2 supercell with eight atomic layers. The dimensions of the surface model were all greater than 11 Å in the surface lateral direction. In the surface normal direction, the slab was separated by a vacuum layer of more than 20 Å, with a total thickness of 36 Å. The total atom number in this surface model was 128. During the structural optimization process, the lower four atomic layers were fixed to simulate bulk effects on the surface, while the coordinates of the upper four atomic layers were completely relaxed to achieve equilibrium. In order to simulate the trace precipitation effect of Ag in Ti-Ta-Ag ternary alloy, one Ag atom was adsorbed onto the surface of (Ti$_{1-x}$Ag$_x$)$_3$Ta ($x = 0$). In order to simulate the corrosion resistance of Ti-Ta-Ag ternary alloy in the application of biomaterials, we analysed Cl$^-$ ions, which are the main component in Hank's solution, as adsorbents. The simulated annealing method was used to determine the exact location of Ag or Cl$^-$ ions on the surface.

The DFT calculation in this work was completed by the CASTEP module in Materials Studio software [28]. Ultra-soft pseudopotential was selected, and the cutoff energy was set to 420 eV. The functional for

exchange-correlation potential was chose as PBEsol in generalized gradient approximation. The BFGS algorithm was selected for structural optimization. Its convergence criteria are set as follows: energy change less than $5.0 \times 10^{-6}$ eV atom$^{-1}$, maximum force less than 0.01 eV Å$^{-1}$, maximum stress less than 0.02 GPa and maximum displacement less than $5.0 \times 10^{-4}$ Å. After the surface models reached the equilibrium state through structural optimization, the total energy and electronic structure were further calculated. The setting for simulated annealing is briefly described as follows: number of cycle was set as 10 with $10^5$ steps per cycle; maximum temperature was set as $10^5$ K and the final temperature was set as 100 K.

# 3. Results and discussion

At the beginning of this section, it is necessary to briefly summarize the corresponding findings in our previous work: (i) Ag atoms can replace Ti or Ta atoms in Ti-Ta binary alloy to form Ti-Ta-Ag ternary alloy, and the probability of occupying Ti atoms is greater. (ii) The alloying and trace precipitation of Ag in Ti-Ta binary alloy are determined by its amount and the sintering temperature of SPS. In other words, when they are below a certain value, the Ag alloying is more obvious; on the contrary, when they are higher than a certain value, trace precipitation of Ag on the Ti-Ta-Ag ternary alloy surface is more obvious. (iii) The mechanical properties of Ti-Ta binary alloy can be enhanced by Ag alloying. (iv) The above phenomena are related to the fact that the 4d$^{10}$ electronic states of Ag have changed from obvious local electronic states to mixed local and non-local electronic states.

For medical materials, the presence of pores will have a very adverse effect on the mechanical properties and corrosion resistance of the alloy. In this work, we first conducted a direct visual observation of the polished Ti-Ta-Ag samples. It was found that the surface of the sample after polishing is very flat and there are no obvious pits, shrinkage and cracks on the macroscopic level, which indicated that the densities of the Ti-Ta-Ag alloy samples should be relatively high. In order to further quantitatively describe the density of Ti-Ta-Ag alloy samples, the relative density ($\rho_r = [\rho_a/\rho_t] \times 100\%$) was measured. The theoretical density ($\rho_t$) of Ti-Ta-Ag alloy is calculated from the composition proportion, while the actual density ($\rho_a$) is measured by Archimedes drainage method. We found that the difference in composition had little impact on the densities, while the SPS sintering temperature had a slight impact. The relative densities ($\rho_r$) of Ti-Ta-Ag alloy samples sintered at 750, 850 and 950°C were about 95.44, 96.84 and 99.12%, respectively. These measurements indicate that the Ti-Ta-Ag alloy prepared in this work has high densities and can meet the requirements of medical human implant materials.

## 3.1. Open-circuit potential testing

As shown in figure 1a, the open-circuit potential ($V_{OP}$) curves of Ti-25Ta-$x$Ag alloy, which contain different Ag content and were treated at different sintering temperatures, showed an increasing trend and gradually tended to be stable with the prolonging of test time (i.e. the longer immersion time of Hank's solution). However, they varied in the range of open-circuit voltage changes over the entire time range and the time required for stabilizing. As shown in figure 1b, the $V_{OP}$ value of the 950°C sintered Ti-25Ta sample and the Ti-25Ta-4.5Ag sample sintered by different temperatures had a relatively large range (both exceeding 64 mV), among which the 950°C sintered Ti-25Ta-4.5Ag sample had the largest range of $V_{OP}$ value, reaching 165.4 mV, and took the longest time to stabilize. However, the $V_{OP}$ values of Ti-25Ta-1.5Ag and Ti-25Ta-1.5Ag samples sintered at different temperatures changed relatively little in the whole time range (less than 32 mV). Among them, the 950°C sintered Ti-25Ta-1.5Ag sample had the smallest range of $V_{OP}$ value, only 10.6 mV, and it needed the shortest time to stabilize. The change of open-circuit potential with time indicates the formation speed of passivation layer on the alloy surface. If passivation occurs on the alloy surface in a corrosive environment, and a stable and dense passivation layer can be formed in a short time, the open-circuit potential can quickly tend to a stable value [14]. That is, the alloy sample can play a stable application function in the corrosive environment. According to this criterion, the Ti-25Ta-1.5Ag sample sintered at 950°C performed best, while Ti-25Ta-4.5Ag sample sintered at 950°C performed worst.

On the other hand, the final stable open-circuit potential values of these samples were also different, as shown in figure 1c. The final stable open-circuit potential values of Ti-25Ta-1.5Ag and Ti-25Ta-3Ag samples sintered at 750°C and Ti-25Ta and Ti-25Ta-4.5Ag samples sintered at 950°C were relatively small (less than −207 mV). The final stable open-circuit voltage of Ti-25Ta-4.5Ag sintered at 750°C,

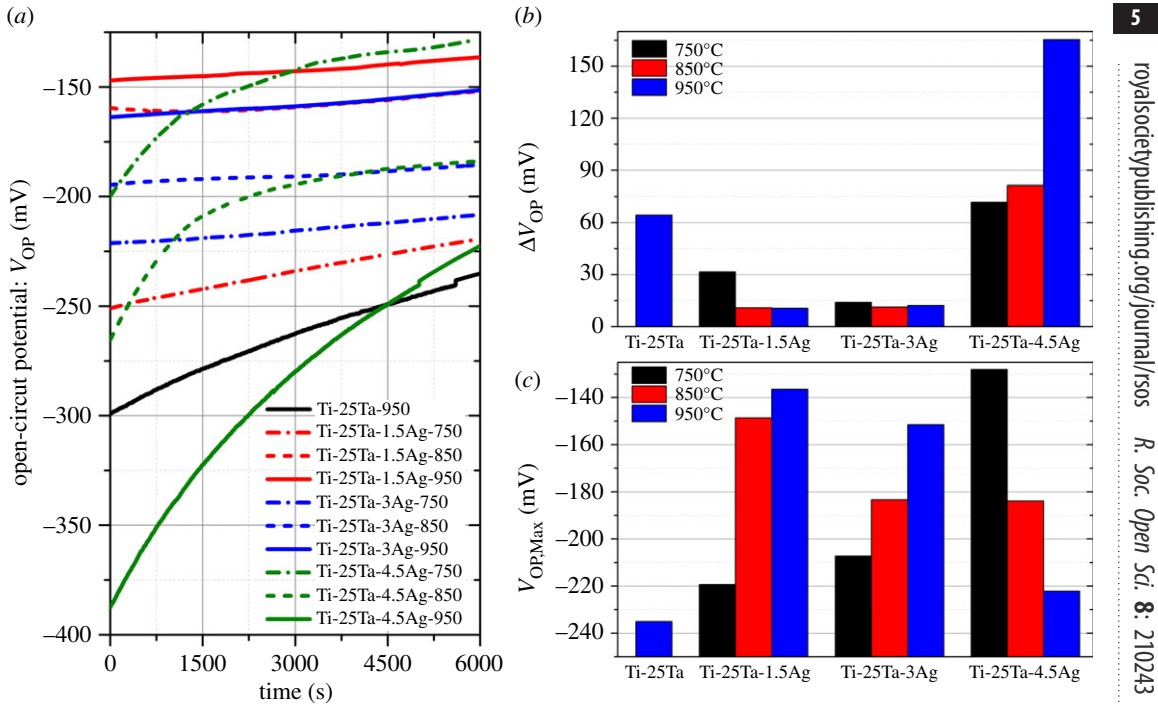

**Figure 1.** (*a*) The open-circuit potential curves, (*b*) the amplitude of open-circuit potential change and (*c*) the final stable open-circuit potential value of Ti-25Ta-*x*Ag alloy, which contain different Ag content and treated at different sintering temperature.

Ti-25Ta-1.5Ag and Ti-25Ta-3Ag sintered at 950°C were relatively large (greater than −150 mV). The final stable open-circuit potential of Ti-25Ta sintered at 950°C was the smallest (−235.0 mV), while that of Ti-25Ta-4.5Ag sintered at 750°C was the largest (−128.0 mV). The higher the open-circuit potential value of the alloy sample, the better the corrosion resistance. According to this criterion, Ag alloying can effectively reduce the corrosion tendency of Ti-Ta binary alloy, showing excellent corrosion resistance.

Through an analysis of the different Ag content and SPS sintering temperature processing of Ti-Ta-Ag ternary alloy samples in Hank's solution of the open-circuit potential test, we found the following rules: (i) Ag alloying can effectively enhance the Ti-Ta binary alloy corrosion resistance, such as Ti-25Ta-1.5Ag and Ti-25Ta-3Ag samples as the SPS sintering temperature increases and Ag alloying degree increases, so the corresponding corrosion resistance is correspondingly increased. (ii) The trace precipitation of Ag is not conducive to the enhancement of the corrosion resistance of Ti-Ta binary alloy. For example, with the increase of SPS sintering temperature, the amount of Ag precipitation of Ti-25Ta-4.5Ag samples gradually increases, so the corresponding corrosion resistance decreases accordingly.

## 3.2. Electrochemical polarization testing

Figure 2 illustrates the electrochemical polarization curves of Ti-25Ta-*x*Ag ternary alloy in Hank's solution. During the cathodic polarization process, all of the current density curves of Ti-25Ta-*x*Ag alloys are decreasing, with the increase of potential; while during the anodic polarization process, all of the current density curves of Ti-25Ta-*x*Ag alloys are increasing, with the increase of potential. Furthermore, they tend to have relatively stable current density. This phenomenon suggests that Ti-25Ta-*x*Ag alloys are undergoing an obvious activation-passivation transition in Hank's solution. In the anodic polarization process, the current densities of Ti-25Ta-1.5Ag and Ti-25Ta-3Ag increased slowly with the increase of potential, while the current densities of Ti-25Ta and Ti-25Ta-4.5Ag increased abruptly with the increase of potential. This was due to the insufficient growth of surface passivation layer thickness, which required a certain current density to compensate.

Table 1 provides the parameters and fitting results of electrochemical polarization curves of Ti-25Ta-*x*Ag ternary alloys in Hank's solution. According to the values of initiating passivation current density and potential ($I_p$ and $E_p$), the higher the degree of Ag alloying, the smaller the absolute value of $I_p$ and $E_p$, and the greater the corresponding polarization resistance. For example, the polarization resistance of Ti-25Ta-1.5Ag and Ti-25Ta-3Ag alloy samples increases with the increase of SPS sintering temperature. On

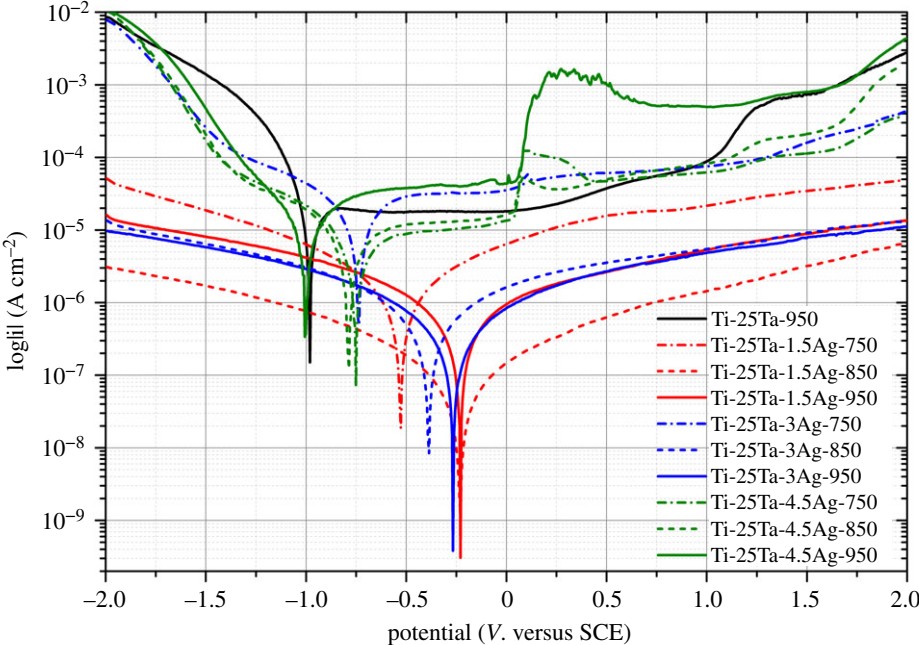

**Figure 2.** Cyclic electrochemical polarization curves of Ti-25Ta-xAg ternary alloys in Hank's solutions.

the contrary, the larger the amount of Ag precipitation, the smaller the polarization resistance of the corresponding sample. For example, the polarization resistance of Ti-25Ta-4.5Ag alloy sample decreases with the increase of SPS sintering temperature. The smaller the initiating passivation current density is (or the greater the polarization resistance is), the easier it is for the alloy sample to enter the passivation state from the activation state, and the higher the stability of the passivation layer is, making it more conducive to the enhancement of corrosion resistance. In this sense, the 950°C SPS sintered Ti-25Ta-1.5Ag and Ti-25Ta-3Ag alloy samples are most conducive to enhance the corrosion resistance.

The self-corrosion current density ($I_{corr}$) indicates the corrosion degree of the alloy sample in Hank's solution. The smaller the $I_{corr}$, the lower the corrosion rate, and the stronger the protection ability of the surface passivation layer to Ti-Ta-Ag alloy. On the other hand, the self-corrosion potential ($E_{corr}$) expresses the degree of corrosion of the alloy sample in Hank's solution. The higher the $E_{corr}$ value of the alloy sample, the smaller its corrosion tendency [29]. Therefore, by comparing the values of $I_{corr}$ and $E_{corr}$ in table 1, it can be concluded that Ag alloying is beneficial to improving the protection ability and corrosion resistance of the surface passivation layer to Ti-Ta-Ag alloy. However, the trace precipitation of Ag is not favourable for this. Finally, the corrosion rate of the alloy sample can be further estimated according to the test results of the self-corrosion current density, as shown in the last column in table 1. Not surprisingly, the corrosion rate of the Ti-25Ta-1.5Ag-950 sample is the smallest, while that of the Ti-25Ta-4.5Ag-950 sample is the largest. This result is consistent with previous open-circuit potential tests.

## 3.3. Electrochemical impedance testing

Figure 3 plots the Nyquist spectra of Ti-25Ta-xAg ternary alloy in Hank's solution. Both Ti-25Ta-1.5Ag and Ti-25Ta-3Ag alloy samples are characterized by a single capacitive reactance arc, without inductive reactance arc, indicating that a passivation layer can be formed on the alloy surface in Hank's solution. However, both Ti-25Ta and Ti-25Ta-4.5Ag alloy samples present divergent capacitive reactance arcs and initial signs of an inductive reactance arc, indicating that it is relatively difficult to form a passivation layer on their surface in Hank's solution. Ti-25Ta-xAg alloy samples with different Ag content and different SPS sintering temperatures have different capacitive reactance arc radii. Ti-25Ta-1.5Ag-950 and Ti-25Ta-3Ag-950 alloy samples have the smallest capacitive reactance arc fitting radius. The smaller the fitting radius of the capacitive reactance arc, the larger the transfer resistance of the material surface, so the larger the impedance and the better the corrosion resistance of the intrinsic passivation layer of the alloy sample. This result proves again that Ag alloying is beneficial to improving the corrosion resistance of Ti-Ta-Ag ternary alloy.

**Table 1.** The parameters and fitting results of electrochemical polarization curves of Ti-25Ta-*x*Ag ternary alloys in Hank's solution.

| sample | $I_p$ ($10^{-7}$ A cm$^{-2}$) | $E_p$ (mV) | $R_p$ (kΩ · cm$^2$) | $I_{corr}$ ($10^{-7}$ A cm$^{-2}$) | $E_{corr}$ (mV) | Corrosion rate ($10^{-5}$ mm ye$^{-1}$ ar) |
|---|---|---|---|---|---|---|
| Ti-25Ta-950 | 1.516 | −981.034 | 0.647 | 13.661 | −981.042 | 4797.000 |
| Ti-25Ta-1.5Ag-750 | 0.189 | −527.152 | 2.794 | 0.716 | −529.541 | 251.385 |
| Ti-25Ta-1.5Ag-850 | 0.012 | −229.063 | 19.072 | 0.041 | −229.360 | 14.330 |
| Ti-25Ta-1.5Ag-950 | 0.003 | −229.061 | 75.337 | 0.007 | −228.750 | 2.430 |
| Ti-25Ta-3Ag-750 | 5.028 | −735.589 | 0.146 | 15.134 | −738.136 | 5314.378 |
| Ti-25Ta-3Ag-850 | 0.084 | −385.065 | 4.609 | 0.415 | −387.926 | 145.754 |
| Ti-25Ta-3Ag-950 | 0.004 | −266.346 | 70.756 | 0.159 | −266.895 | 55.985 |
| Ti-25Ta-4.5Ag-750 | 0.717 | −751.464 | 1.048 | 3.946 | −750.647 | 1385.680 |
| Ti-25Ta-4.5Ag-850 | 1.255 | −785.321 | 0.626 | 3.482 | −788.592 | 1222.829 |
| Ti-25Ta-4.5Ag-950 | 3.386 | −1005.636 | 0.297 | 26.570 | −1001.383 | 9330.044 |

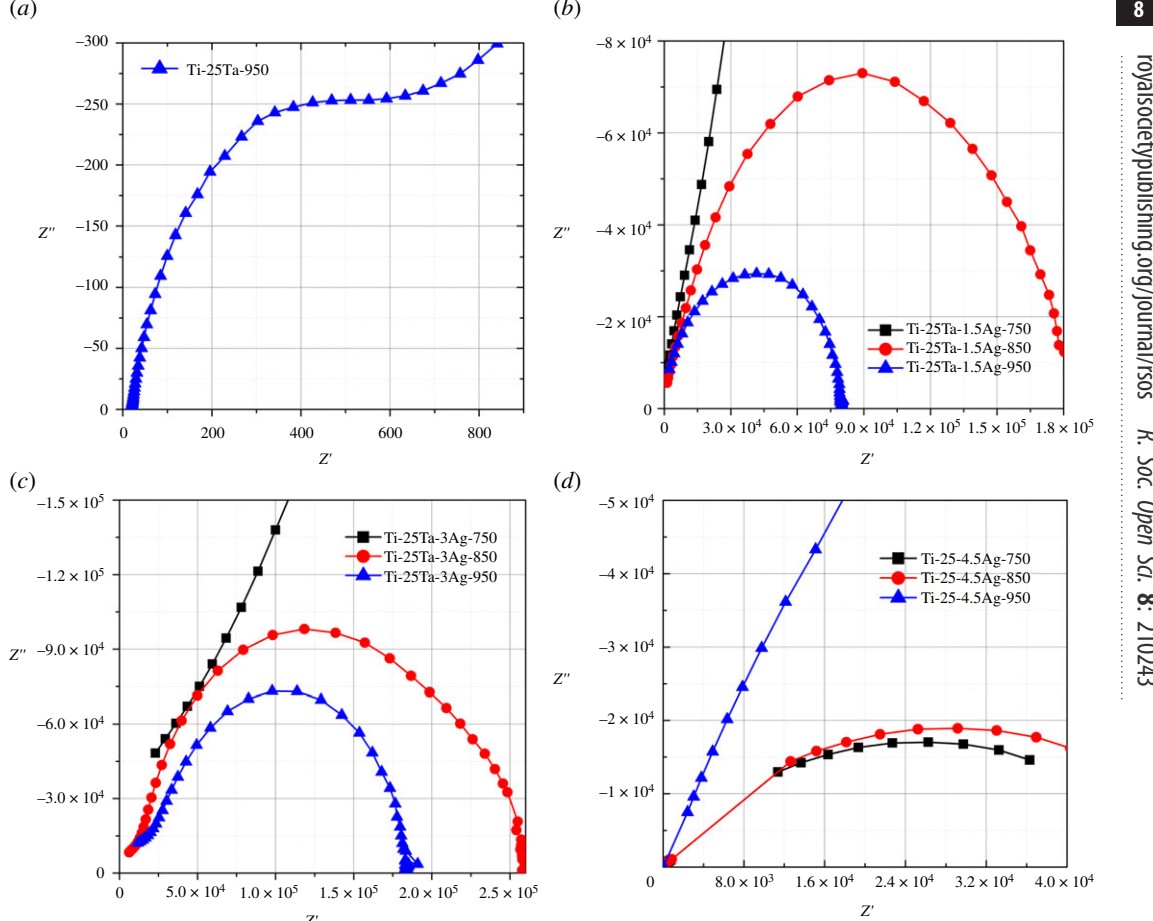

**Figure 3.** Nyquist spectra of Ti-25Ta-xAg ternary alloy samples in Hank's solutions: (a) Ti-25Ta sample; (b) Ti-25Ta-1.5Ag samples, (c) Ti-25Ta-3Ag samples and (d) Ti-25Ta-4.5Ag samples.

Figures 4 and 5 show the Bode impedance spectra and phase-angle spectra of Ti-25Ta-xAg ternary alloys in Hank's solution. Both Ti-25Ta-1.5Ag and Ti-25Ta-3Ag alloy samples exhibit high impedance modulus in a wide range of low frequencies, and the phase-angle tends to 0°, showing pure resistance characteristics. In the medium- and high-frequency range, the impedance spectra are linear, in which the slope is about −1 and the phase-angle reaches 70–75°, indicating that the passivation layer has high capacitance behaviour. Thus, for these alloy samples, the dielectric properties of the passivation layer remain stable with time. The impedance curves of these alloy samples are smooth, without obvious fluctuation characteristics, indicating that the passivation layer generated on the alloy surface is complete, uniform and without defects. Therefore, the corrosion process of these alloy samples is uniform, and there is no pitting corrosion phenomenon. However, with the increase of frequency, the impedance modulus of Ti-25Ta and Ti-25Ta-4.5Ag alloy samples continues to decrease. Over a wide frequency range, the impedance spectrum is linear and finally becomes stable. These test results show that the high resistance of the passivation layer of these alloy samples continues to decline, and the dielectric properties of the passivation layer continue to decrease with the increase of time and then remain stable. Furthermore, the impedance curves of Ti-25Ta and Ti-25Ta-4.5Ag alloy samples have significantly reduced the fluctuation, which indicates that the passivation layer generated on the surface of these alloy samples is incomplete or uneven, and there are some defects. Therefore, their corrosion process is uneven and pitting corrosion phenomenon appears [30–32].

In order to conduct parameter analysis and data fitting for electrochemical impedance spectrum data, it is necessary to establish an equivalent circuit model of the double-layer passivation layer, as shown in the inner illustration of figure 5a. $R_s$ represents the contact resistance between solution and work electrode, $CPE_P$ and $R_P$ represent the constant phase-angle element and resistance of the outer porous layer, respectively, and $CPE_b$ and $R_b$ represent the constant phase-angle element and resistance of the inner dense layer, respectively. The fitting results are shown in table 2, and the error $\chi^2$ values of each fitting parameter are all less than $10^{-3}$, indicating that the fitting data are in good agreement with the

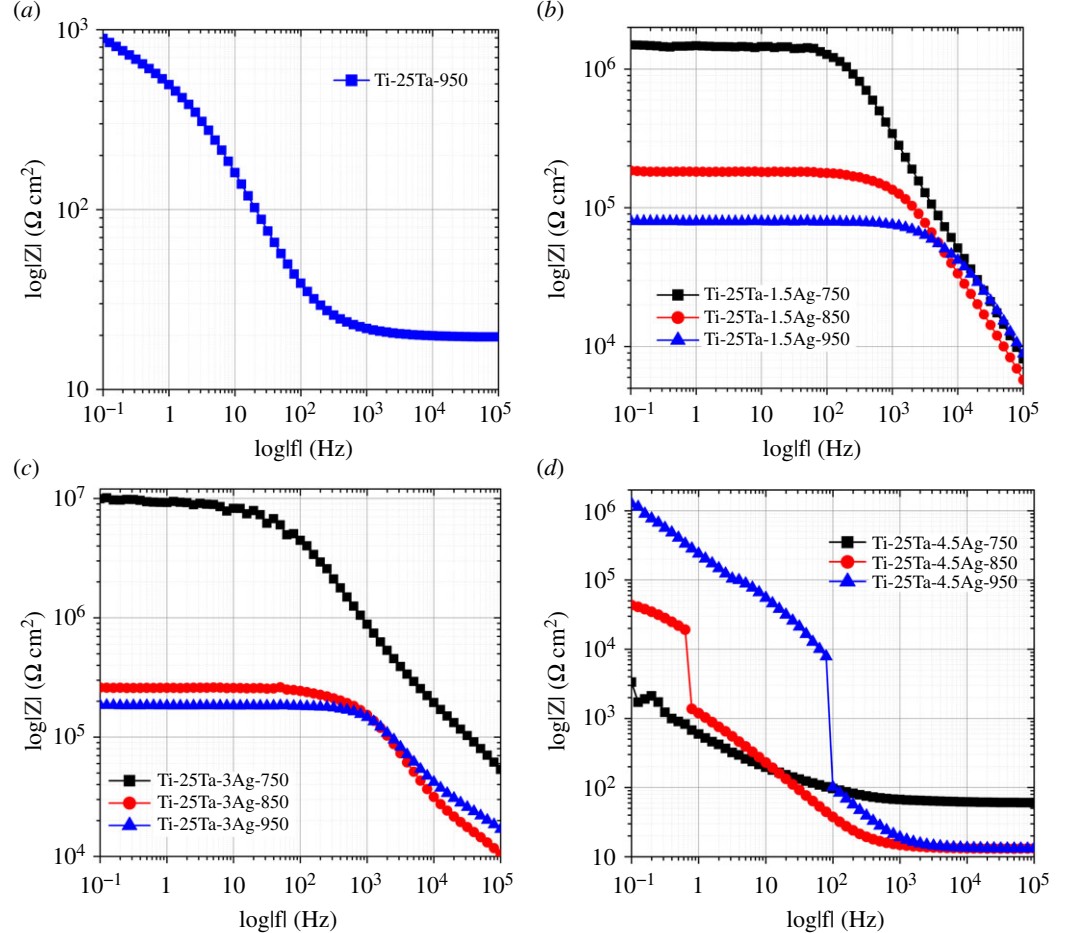

**Figure 4.** Bode impedance spectra of Ti-25Ta-*x*Ag ternary alloy samples in Hank's solution: (*a*) Ti-25Ta sample, (*b*) Ti-25Ta-1.5Ag samples, (*c*) Ti-25Ta-3Ag samples and (*d*) Ti-25Ta-4.5Ag samples.

testing values. In both Ti-25Ta-1.5Ag and Ti-25Ta-3Ag alloy samples, the resistance $R_b$ of the inner dense layer is relatively high and is much higher than that of the outer porous layer $R_p$. Moreover, these fitted resistance values are all larger than those of Ti-Ta alloy sample, indicating that Ag alloying can help to increase the thickness of passivation layer. These results indicate that the corrosion resistance of these alloy samples mainly depends on the inner dense layer, which is the main source of corrosion resistance of Ti-Ta-Ag alloy samples. The values of $n_2$ in the inner dense layer of these alloy samples are all close to 1, indicating that this layer is an ideal capacitor. The $n_1$ values of the outer porous layer are slightly lower, indicating that the electrolyte enters the inner pore and is forming a new surface layer. In addition, the variation of $R_s$ values of Ti-25Ta-1.5Ag and Ti-25Ta-3Ag alloy samples, as well as the variation of $R_s$ values relative to Ti-25Ta alloy samples, indicates that Ag alloying is beneficial for reducing the contact resistance of the solid–liquid interface.

Although the corresponding fitting values of Ti-25Ta and Ti-25Ta-4.5Ag alloy samples also show a similar rule to the above phenomenon, their $R_b$ and $R_p$ values are far lower than those of the other two alloy samples. Therefore, Ag trace precipitation will reduce the thickness of the passivation layer on the alloy surface, which is not conducive to enhancing the corrosion resistance of the alloy samples. In addition, it is worth noting that although the protection of the outer porous layer to the alloy samples is not as obvious as that of the inner dense layer, the diffusion and binding processes of ions and molecules in body fluid occur in the pores of the outer porous layer. Moreover, the porous structure facilitates the adhesion of bone cells. The above two reasons make the alloy samples have better bioactivity and biocompatibility, and become good implant materials that can be used in human body.

## 3.4. Surface adsorption and electronic structure

In the above corrosion test, we found that for Ti-Ta-Ag ternary alloy, Ag alloying is conducive to its corrosion resistance improvement, but the trace precipitation of Ag is not conducive to its corrosion

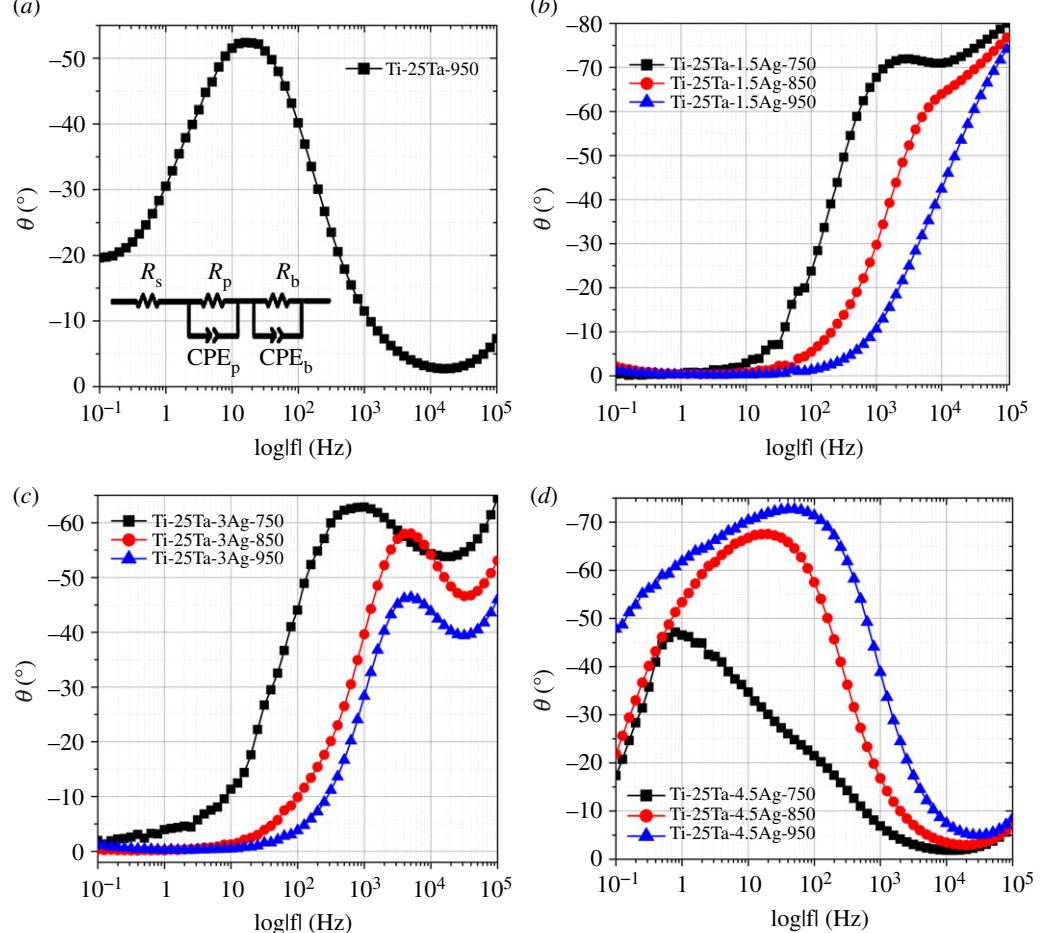

**Figure 5.** Bode phase-angle spectra of Ti-25Ta-*x*Ag ternary alloy samples in Hank's solution: (*a*) Ti-25Ta sample, (*b*) Ti-25Ta-1.5Ag samples, (*c*) Ti-25Ta-3Ag samples and (*d*) Ti-25Ta-4.5Ag samples. The inset of (*a*) is the equivalent circuit model of double-layer passivation layer.

resistance improvement. In order to further analyse the internal physical mechanism behind these experimental phenomena, the surface adsorption and corresponding electronic structure of Ti-Ta-Ag ternary alloy were preliminarily analysed by the DFT theory calculation method. Therefore, on the basis of our previous work, we constructed three surface models, namely (001) surface of Ti-25Ta, Ti-25Ta-Ag and Ag adsorbed Ti-25Ta (labelled as 'Ag@Ti-25Ta'), which, respectively, represent Ti-Ta binary alloy, Ti-Ta-Ag ternary alloy and Ti-Ta binary alloy with Ag trace precipitation. On the basis of the structural optimization of the three surface models, the most stable adsorption position of Cl atom was determined by the simulated annealing method. Then, the structure of the surface adsorption model was optimized again, and the microstructure parameters of Cl$^-$ ions adsorption on the alloy surface were obtained. Finally, the total energy and electronic structure of all models were calculated.

After surface simulated annealing and structure optimization, surface stable adsorption of Cl atom is shown in figure 6*a*. In the case of Ti-25Ta-(001) surface, Cl$^-$ ions are stably adsorbed directly above the centre of the triangle with one Ta atom and two Ti atoms as the vertices. In this optimized model, the length of Ti-Cl bond and Ta-Cl bond are very close, and their average bond length is 2.506 Å. In the case of Ti-25Ta-Ag-(001) surface, the stable adsorption configuration of Cl atom is similar to that on the Ti-25Ta-(001) surface. The stable adsorption site is still directly above the centre of the triangle with one Ta atom and two Ti atoms as its vertices. The average bond length of Ti-Cl and Ta-Cl bonds is 2.439 Å. In other words, on the surface of Ti-Ta-Ag ternary alloy, the stable adsorption position of Cl atom is far from the surface Ag atom, and there is no direct interaction between them. However, Ag alloying still has an effect on the adsorption of Cl atom, increasing its adsorption energy by about 0.326 eV atom$^{-1}$ and reducing the average bond length by about 0.067 Å, as shown in figure 6*b*. The results show that the interaction between Cl$^-$ ions in Hank's solution and the surface of Ti-Ta-Ag ternary alloy is enhanced by Ag alloying.

**Table 2.** Electrochemical impedance spectroscopy fitting values of alloy samples in Hank's solution.

| sample | $CPE_b$ (S sn cm$^{-2}$) | $n_2$ | $R_b$ ($\Omega$ cm$^2$) | $CPE_p$ (S sn/cm$^2$) | $n_1$ | $R_p$ ($\Omega$ cm$^2$) | $R_s$ ($\Omega$ cm$^2$) | $\chi^2$ |
|---|---|---|---|---|---|---|---|---|
| Ti-25Ta-950 | $2.8 \times 10^{-4}$ | 0.81 | 816.8 | 1.67 | 0.67 | 424.7 | 19.5 | $9.0 \times 10^{-4}$ |
| Ti-25Ta-1.5Ag-750 | $6.6 \times 10^{-11}$ | 0.92 | $3.3 \times 10^6$ | $1.5 \times 10^{-10}$ | 0.91 | $6.4 \times 10^5$ | 18.7 | $6.8 \times 10^{-4}$ |
| Ti-25Ta-1.5Ag-850 | $3.2 \times 10^{-12}$ | 0.95 | $8.2 \times 10^6$ | $1.3 \times 10^{-10}$ | 0.83 | $2.2 \times 10^5$ | 15.5 | $2.5 \times 10^{-4}$ |
| Ti-25Ta-1.5Ag-950 | $6.7 \times 10^{-11}$ | 0.93 | $1.7 \times 10^7$ | $1.7 \times 10^{-10}$ | 0.89 | $2.3 \times 10^5$ | 17.4 | $2.1 \times 10^{-4}$ |
| Ti-25Ta-3Ag-750 | $6.9 \times 10^{-11}$ | 0.93 | $1.2 \times 10^7$ | $5.2 \times 10^{-10}$ | 0.83 | $1.5 \times 10^5$ | 16.7 | $7.4 \times 10^{-4}$ |
| Ti-25Ta-3Ag-850 | $2.1 \times 10^{-11}$ | 0.95 | $3.4 \times 10^6$ | $4.5 \times 10^{-10}$ | 0.87 | $2.1 \times 10^5$ | 15.3 | $1.8 \times 10^{-4}$ |
| Ti-25Ta-3Ag-950 | $1.4 \times 10^{-11}$ | 0.98 | $4.1 \times 10^6$ | $2.3 \times 10^{-10}$ | 0.89 | $2.6 \times 10^5$ | 14.8 | $2.1 \times 10^{-4}$ |
| Ti-25Ta-4.5Ag-750 | $8.34 \times 10^{-7}$ | 0.96 | $1.50 \times 10^6$ | $4.47 \times 10^{-6}$ | 0.89 | $7.80 \times 10^4$ | 20.49 | $4.72 \times 10^{-4}$ |
| Ti-25Ta-4.5Ag-850 | $1.13 \times 10^{-5}$ | 0.95 | $4.05 \times 10^4$ | $2.22 \times 10^{-5}$ | 0.54 | 116.6 | 27.95 | $3.07 \times 10^{-4}$ |
| Ti-25Ta-4.5Ag-950 | $5.71 \times 10^{-5}$ | 0.89 | $2.147 \times 10^3$ | $3.84 \times 10^{-4}$ | 0.57 | 412.5 | 59.55 | $1.19 \times 10^{-4}$ |

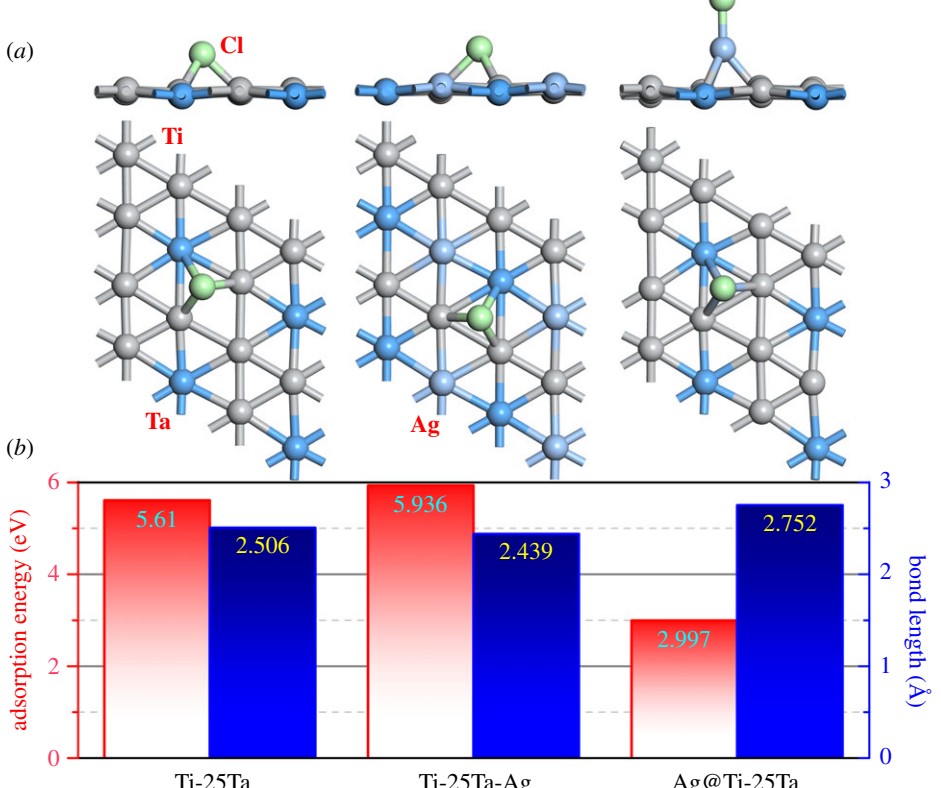

**Figure 6.** (a) The top and side view of stable adsorption of Cl atom onto the Ti-Ta binary alloy surface, Ti-Ta-Ag ternary surface and Ti-Ta binary alloy surface with precipitated Ag atom; (b) the corresponding adsorption energy and bond length with the surface.

Ag atoms precipitate from Ti-Ta-Ag ternary alloy and adsorb on the surface. The stable adsorption configuration of precipitated Ag atom is similar to that of Cl atom on the surface of Ti-25Ta. On the basis of this, Cl atom will be directly adsorbed to precipitated Ag atom on the surface, and the corresponding adsorption energy is greatly reduced to 2.997 eV atom$^{-1}$. In the case of Ag@Ti-25Ta-(001) surface, the average bond length of Ti-Ag and Ta-Ag on the surface is 2.752 Å, and the average bond length of Ag-Cl is 2.310 Å. The former is larger than the corresponding bond length on Ti-25Ta-(001) surface and Ti-25Ta-Ag-(001) surface, while the latter is close to the bond length of isolated AgCl molecule (2.264 Å). These calculated results show that Cl$^-$ ions in Hank's solution are more likely to combine with precipitated Ag atoms on the surface of Ti-Ta-Ag ternary alloy, forming an adsorption configuration similar to that of isolated AgCl molecules. Moreover, this surface adsorption interaction is weaker than the first two cases. In summary, these calculated results show that the Ag alloying enhances the interaction between Cl$^-$ ions and the surface of Ti-Ta-Ag ternary alloy in Hank's solution, while the Ag trace precipitation weakens the interaction between Cl$^-$ ions and the surface of Ti-Ta-Ag ternary alloy in Hank's solution.

In order to further analyse the underlying mechanism of the above surface interaction changes, the surface electronic structure before and after Cl atom adsorption is provided in figure 7. As shown in figure 7a, the comparison of the three surface electronic structures reveals that the 4d electronic states of the alloyed Ag atom are in a deeper energy range and exhibit obvious localized characteristics. By contrast, the 4d electron states of precipitated Ag atoms on the surface increased in energy position and showed some non-localized characteristics. The electronic structure of the latter is very similar to that of $(Ti_{1-x}Ag_x)_3Ta$ solid solution when $x > 0.8$ [27]. The surface electronic structure after adsorption of Cl atom is shown in figure 7b. On the Ti-25Ta-(001) surface, the Cl-3p states show obvious localized characteristics. That is, although it bonds with Ti and Ta atoms on the surface, the interaction between them is not strong due to the long bond length. Therefore, the corrosion resistance of Ti-Ta binary alloy in Hank's solution is poor. On the Ti-25Ta-Ag(001) surface, although the Cl-3p states still maintain the localized characteristics, it has been obviously weakened. Moreover, since the Cl-3p states just overlap with the Ag-4d states in energy, the interaction between adsorbed Cl atom and surface atoms is further enhanced. Therefore, Ag alloying on the surface of Ti-Ta-Ag ternary alloy

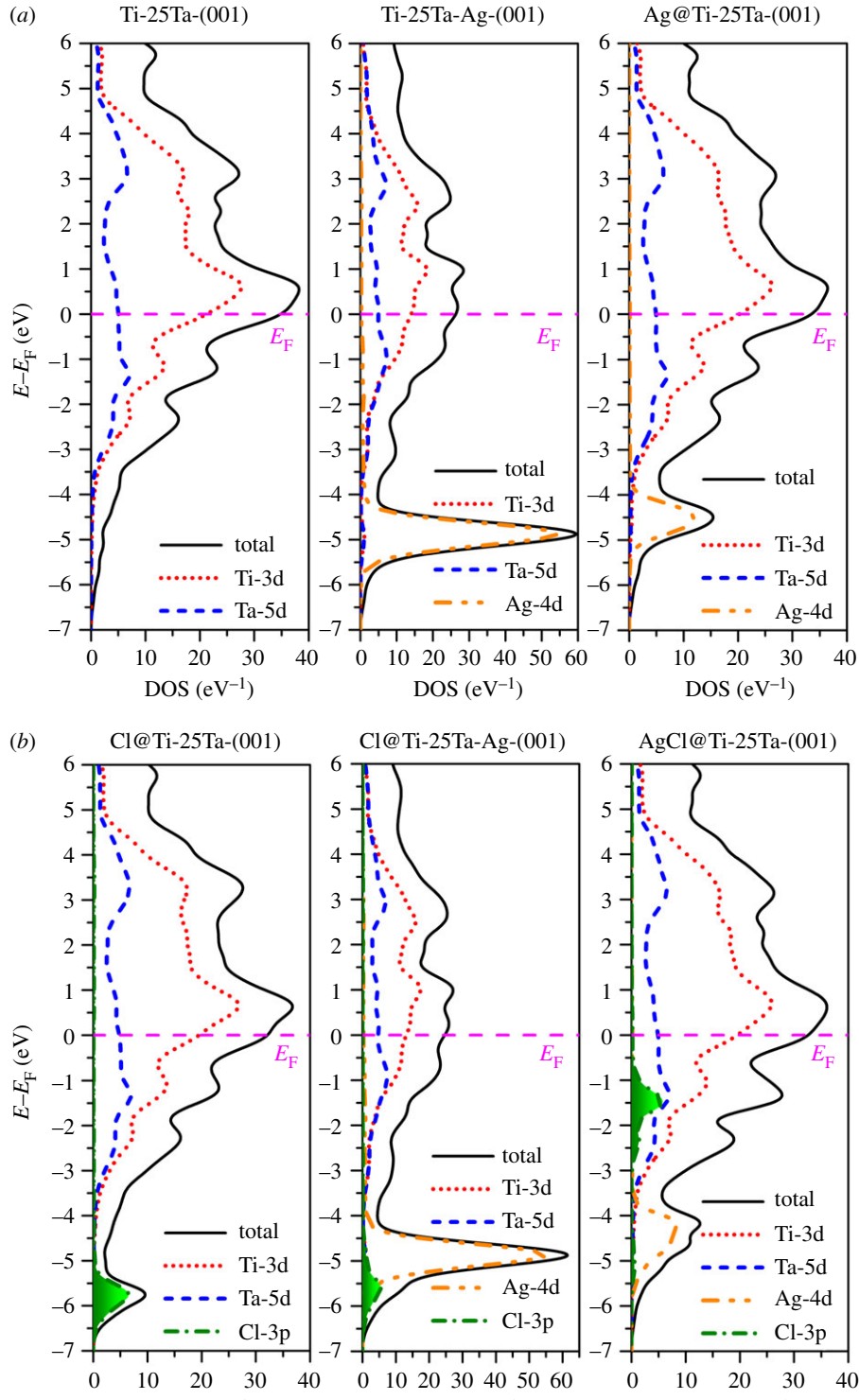

**Figure 7.** The density of states of before (*a*) and after (*b*) Cl atom adsorption onto Ti-Ta binary alloy surface, Ti-Ta-Ag ternary surface and Ti-Ta binary alloy surface with precipitated Ag atom.

can significantly enhance its corrosion resistance in Hank's solution. On the Ag@Ti-25Ta-(001) surface, the energy range occupied by the Cl-3p states increases obviously, and it is separated from the Ag-4d states in energy. However, the adsorbed Cl atom is directly bonded with the precipitated Ag atom on this surface. This calculated result implies that the Ag-Cl bond is an ionic bond, and they interact with each other through cation and anion. This is a relatively strong interaction, so that AgCl becomes a similar molecular configuration, as a whole adsorbed onto the surface of Ti-Ta-Ag alloy. The bond length between the latter and the surface is longer and the interaction is weaker. The combination of

these two effects makes the Ag trace precipitation on the surface of Ti-Ta-Ag alloy weaken its corrosion resistance in Hank's solution.

## 4. Conclusion

In order to further investigate the effect of Ag trace precipitation in Ti-Ta-Ag ternary alloys on their performance as medical implant materials, this work systematically investigated the corrosion behaviour of samples prepared under different SPS process conditions in artificial simulated body fluids. Ag alloying can promote the formation of a uniform, complete, dense, stable and thick passivation layer on the surface of Ti-Ta-Ag ternary alloy, which makes Ti-Ta-Ag ternary alloy uniformly corroded in Hank's solution without pitting. In addition, Ag alloying can effectively reduce the contact resistance of solid–liquid interface. These effects significantly enhance the corrosion resistance of Ti-Ta-Ag ternary alloy in Hank's solution. However, if there is a small amount of Ag precipitation on the surface of Ti-Ta-Ag ternary alloy, the trace precipitation of Ag plays the opposite role to the above effect. The internal mechanism of the above phenomena is investigated by DFT calculation: on the Ag alloyed surface, $Cl^-$ ions in Hank's solution are more strongly bound to the surface atoms, which is more conducive to the formation and stability of a dense passivation layer; while on the surface of Ag trace precipitation, $Cl^-$ ions are closely bonded to the precipitated Ag atoms, forming isolated adsorption molecules similar to AgCl, which is adverse to the formation and stability of the passivation layer. Therefore, in order to improve the corrosion resistance of Ti-Ta-Ag ternary alloys, trace precipitation of Ag needs to be avoided as much as possible in the preparation.

Data accessibility. The datasets supporting this article have been uploaded as part of the electronic supplementary material.

Authors' contributions. J.M.Z., J.D.Y., X.H.C. and Y.H.L. carried out the experiments; Z.Y.Z. carried out the DFT calculation. J.M.Z. wrote the manuscript; Z.Y.Z. and Q.H.C. revised the manuscript.

Competing interests. The authors declare no competing financial interests.

Funding. The authors acknowledge financial support from the National Natural Science Foundation of China (grant no. 11964015).

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
