## [Peer Review File · Royal Society Open Science]

Review History

RSOS-210243.R0 (Original submission)

Review form: Reviewer 1

Is the manuscript scientifically sound in its present form?

No

Are the interpretations and conclusions justified by the results?

No

Is the language acceptable?

No

Do you have any ethical concerns with this paper?

No

Have you any concerns about statistical analyses in this paper?

No

Recommendation?

Major revision is needed (please make suggestions in comments)

Comments to the Author(s)

See attached (Appendix A).

Review form: Reviewer 2

Is the manuscript scientifically sound in its present form?

Yes

Are the interpretations and conclusions justified by the results?

Yes

Is the language acceptable?

Yes

Do you have any ethical concerns with this paper?

Yes

Have you any concerns about statistical analyses in this paper?

No

Recommendation?

Accept as is

Comments to the Author(s)

Dear authors, the attempt and the results are somewhat interests of ternary alloys for corrosion resistance application for engineering and automobiles. This may be okay; however, there are several data's need to be addressed such as surface morphologies of your materials. Anyhow, I accept this work at presently in this esteemed Journal.

Decision letter (RSOS-210243.R0)

Dear Professor Zhao:

Title: Effect of Ag alloying and trace precipitation on corrosion resistance of Ti-Ta-Ag ternary alloy

Manuscript ID: RSOS-210243

The editor assigned to your manuscript has now received comments from reviewers. We would like you to revise your paper in accordance with the referee and Subject Editor suggestions which

can be found below (not including confidential reports to the Editor). Please note this decision does not guarantee eventual acceptance.

Please submit your revised paper before 23-May-2021. Please note that the revision deadline will expire at 00.00am on this date. If we do not hear from you within this time then it will be assumed that the paper has been withdrawn. In exceptional circumstances, extensions may be possible if agreed with the Editorial Office in advance. We do not allow multiple rounds of revision so we urge you to make every effort to fully address all of the comments at this stage. If deemed necessary by the Editors, your manuscript will be sent back to one or more of the original reviewers for assessment. If the original reviewers are not available we may invite new reviewers.

On behalf of the Subject Editor Professor Anthony Stace and the Associate Editor Dr Darren Walsh.

RSC Associate Editor:
Comments to the Author:
(There are no comments.)

RSC Subject Editor:
Comments to the Author:
(There are no comments.)

Reviewers' Comments to Author:

Reviewer: 1

Comments to the Author(s)

See attached

Reviewer: 2

Comments to the Author(s)

Dear authors, the attempt and the results are somewhat interests of ternary alloys for corrosion resistance application for engineering and automobiles. This may be okay; however, there are several data's need to be addressed such as surface morphologies of your materials. Anyhow, I accept this work at presently in this esteemed Journal.

Author's Response to Decision Letter for (RSOS-210243.R0)

See Appendix B.

RSOS-210243.R1 (Revision)

Review form: Reviewer 1

Is the manuscript scientifically sound in its present form?

Yes

Are the interpretations and conclusions justified by the results?

Yes

Is the language acceptable?

Yes

Do you have any ethical concerns with this paper?

No

Have you any concerns about statistical analyses in this paper?

No

Recommendation?

Accept as is

Comments to the Author(s)

The authors have satisfactorily responded to most of my comments and made the necessary changes to the manuscript. The manuscript is accepted for publication.

Decision letter (RSOS-210243.R1)

Dear Professor Zhao:

Title: Effect of Ag alloying and trace precipitation on corrosion resistance of Ti-Ta-Ag ternary alloy

Manuscript ID: RSOS-210243.R1

It is a pleasure to accept your manuscript in its current form for publication in Royal Society Open Science. The chemistry content of Royal Society Open Science is published in collaboration with the Royal Society of Chemistry.

Yours sincerely,
Dr Ellis Wilde
Publishing Editor, Journals

On behalf of the Subject Editor Professor Anthony Stace and the Associate Editor Dr Darren Walsh.

RSC Subject Editor
Comments to the Author:
(There are no comments.)

RSC Associate Editor
Comments to the Author:
(There are no comments.)

Reviewer(s)' Comments to Author:
Reviewer: 1

Comments to the Author(s)

The authors have satisfactorily responded to most of my comments and made the necessary changes to the manuscript. The manuscript is accepted for publication.

Appendix A

Title “Effect of Ag alloying and trace precipitation on corrosion resistance of Ti-Ta-Ag ternary alloy”

Journal: *Royal Society Open Science*

Manuscript ID: RSOS-210243

The paper under review deals with a biomedical material Ti-Ta bearing Ag. The experimental results presented in the work suggest that the Ag alloying has coupled effect on the corrosion resistance of the studied material. The authors also analyzed the surface adsorption and corresponding electronic structure of Ti-Ta-Ag ternary alloy by DFT theory calculation method. However, some comments should be considered before the possible acceptance for publication.

Reviewer's comments:

1#Abstract is poorly written, often lacks important information. Most findings should be presented in the Abstract. The abstract section should be re-written with showing the additions of Ag (0-4.5) and (aging temperatures) and the most findings related to the effect of Ag alloying on the corrosion behavior of the studied material.

2#Remove the first three lines from the abstract.

3#Change “Cl atoms” to “Cl- ions”

4#Change 2. Method details to 2. Experimental procedures.

5#Please add the working sample dimensions.

6#Change “Corrosion resistance testing” to “Electrochemical behavior”:

-Section 2.1 should be re-written to be more concise and informative. Show type of the used electrochemical cell, three-electrode compartment glass cell

7#The characteristics of the porosity in the fabricated sample by SPS (i.e. material density) should be reported

8# What would be the cathodic and anodic reactions associated with the polarization?

9# The conclusions should be refined to be more concise and informative.

10# The presentation of this work should be further improved. Several sentences are not understood. A native English speaker is required.

Appendix B

Title “Effect of Ag alloying and trace precipitation on corrosion resistance of Ti-Ta-Ag ternary alloy”

Journal: *Royal Society Open Science*

Manuscript ID: RSOS-210243

Dear Dr. Ellis Wilde and Reviewers,

Thank you for your e-mail and for the reviewer’s comments concerning above manuscript, and giving us another opportunity to revise it. Those suggestions are all valuable and very helpful for revising and improving our manuscript, as well as the important guiding significance to our researches. We have studied comments carefully and have made correction, which we hope to meet with approval. The main corrections in the paper and the responds to the reviewer’s comments are point-by-point listed as flowing.

We have tried our best to revise this manuscript according to the comments. Please find attached the revised manuscript, which we would like to resubmit for your kind consideration. We would like to express our great appreciation to you and reviewers for suggestions on our manuscript.

Looking forward to hearing from you.

Thank you and best regards.

Yours sincerely,

Zong-Yan Zhao, Ph. D., Prof.

For Reviewer 1

Comment 0:

The paper under review deals with a biomedical material Ti-Ta bearing Ag. The experimental results presented in the work suggest that the Ag alloying has coupled effect on the corrosion resistance of the studied material. The authors also analyzed the surface adsorption and corresponding electronic structure of Ti-Ta-Ag ternary alloy by DFT theory calculation method. However, some comments should be considered before the possible acceptance for publication.

Authors reply:

Thank you very much for your positive affirmation and support of our manuscript. In accordance with your suggestions, we have revised the manuscript, and hope it can continue to get your support and meet the publishing standards.

Comment 1:

Abstract is poorly written, often lacks important information. Most findings should be presented in the Abstract. The abstract section should be re-written with showing the additions of Ag (0-4.5) and (aging temperatures) and the most findings related to the effect of Ag alloying on the corrosion behavior of the studied material.

Authors reply:

Thanks very much for this suggestion. This is very helpful suggestion. According to this suggestion, we have re-write the Abstract, as following (by the way, Abstracts are limited to 200 words in *Royal Society Open Science*):

This work systematically analyzed the electrochemical and corrosion behavior of Ti-Ta-Ag ternary alloy samples in Hank's solution. For the samples with 1.5% and 3% Ag content, the sintering temperature increased from 750 to 950 °C, and the corresponding corrosion resistance increased by 100 times due to the increased alloying of Ag; while for the sample with 4.5% Ag content, the sintering temperature increased from 750 to 950 °C, and the corresponding corrosion resistance decreased by 6 times due to the increased precipitation of Ag. These testing prove that the Ag alloying is beneficial to the enhancement of the corrosion resistance of Ti-Ta-Ag ternary alloy, but the Ag trace precipitation has the opposite effect. A series of electrochemical characterizations and density functional theory calculations explain the mechanism of the above phenomenon. Ag alloying can promote the formation of uniform, complete, dense, stable and thick passivation layer on the surface of Ti-Ta-Ag ternary alloy, which makes Ti-Ta-Ag ternary alloy uniformly corroded without pitting. In addition, Ag alloying can effectively reduce the contact resistance of solid-liquid interface. However, the trace precipitation of Ag just plays the opposite role to the above effect.

Comment 2:

Remove the first three lines from the abstract.

Authors reply:

Thanks for this suggestion. These sentences have been removed in the re-write Abstract.

Comment 3:

Change “Cl atoms” to “Cl- ions”

Authors reply:

Thanks for this suggestion. All the "Cl atoms" words have been replaced by the "Cl⁻ ions" words.

Comment 4:

Change 2. Method details to 2. Experimental procedures.

Authors reply:

Thanks for this suggestion. This time, we added the following details:

In the last paragraph of Page 3: “The raw materials used for the preparation of samples were Ti, Ta and Ag metal powders with purity greater than 99.5%, and their particle sizes were ≤ 55 , ≤ 30 , 7-10 μm , respectively. Firstly, Ti, Ta and Ag metal powders were respectively weighed according to the mass fraction of (75-x)%, 25% and x% ($x = 0, 1.5, 3$ and 4.5). The weighed metal powders were mixed with ZrO_2 as ball milling medium (the ratio of ball to material was maintained at 3.5:1), and then put into the vacuum ball milling tank. The vacuum ball mill tank was sealed and vacuumed to ~ 10 Pa; and then filled with argon as the protection gas. Then the mixed metal powders were ball milled at 150 RPM for 20 hours. Finally, the fully mixed ball-milled metal powders were loaded into the graphite mold ($\text{Ø}20$ mm \times 50 mm), and sintered in the SPS furnace. Sintering process parameters were set as follows: the temperature was raised to the desired temperatures (750, 850, and 950°C) at a heating rate of 90°C/min, the axial pressure of 30 MPa was continuously applied in the sintering process, the vacuum degree of the system was 2-8 Pa, and the sintering time was 5 minutes. After the pulse current is turned off, the samples are quickly cooled to room temperature in the furnace and taken out for subsequent testing.”

Comment 5:

Please add the working sample dimensions.

Authors reply:

Thanks for this suggestion. In the revised manuscript, we added the following expression:

In the second paragraph of Page 4: “In this work, the working electrodes were wafers with a diameter of about 20 mm.”

Comment 6:

Change “Corrosion resistance testing” to “Electrochemical behavior”: -Section 2.1 should be re-written to be more concise and informative. Show type of the used electrochemical cell, three-electrode compartment glass cell.

Authors reply:

Thanks for this suggestion. We have changed the title of this subsection as “Electrochemical behavior”, and added the following information of electrochemical testing:

In the last paragraph of Page 4: “Three electrode comparison method was used in the corrosion experiment: saturated calomel electrode (SCE) was selected as the reference electrode, platinum electrode as the auxiliary electrode, and Ti-Ta-Ag alloy sample as the working electrode. After the above three electrodes were installed in a fixed position, they were put into a quartz electrolytic cell filled with artificial body fluids.”

Comment 7:

The characteristics of the porosity in the fabricated sample by SPS (i.e. material density) should be reported.

Authors reply:

Thanks for this suggestion. In the revised manuscript, we added the following paragraph.

The last paragraph in Page 6: “For medical materials, the presence of pores will have a very adverse effect on the mechanical properties and corrosion resistance of the alloy. In this work, we first conducted direct visual observation of the polished Ti-Ta-Ag samples. It was found that the surface of the sample after polishing is very flat and there are no obvious pits, shrinkage and cracks on the macroscopic level, which indicated that the densities of the Ti-Ta-Ag alloy samples should be relatively high. In order to further quantitatively describe the density of Ti-Ta-Ag alloy samples, the relative density ($\rho_r = [\rho_a/\rho_t] \times 100\%$) was measured. The theoretical density (ρ_t) of Ti-Ta-Ag alloy is calculated from the composition proportion, while the actual density (ρ_a) is measured by Archimedes drainage method. We found that the difference in composition had little impact on the densities, while the SPS sintering temperature had a slight impact on it. The relative densities (ρ_r) of Ti-Ta-Ag alloy samples sintered at 750, 850, and 950°C were about 95.44, 96.84, and 99.12%, respectively. These measurements indicate that the Ti-Ta-Ag alloy prepared in this work has high densities and can meet the requirements of medical human implant materials.”

Comment 8:

What would be the cathodic and anodic reactions associated with the polarization?

Authors reply:

In the polarization process of electrochemical corrosion, oxidation reaction of chloride ions, electrochemical dissolution reactions and ion precipitation reactions occur at the anode, and the reaction expressions are as follows:

The reduction reaction of dissolved oxygen occurs at the cathode with the following reaction expression:

**Comment 9:**

The conclusions should be refined to be more concise and informative.

Authors reply:

Thanks very much for this suggestion. This time, we re-write the conclusion as following:

In order to further investigate the effect of Ag trace precipitation in Ti-Ta-Ag ternary alloys on their performance as medical implant materials, this work systematically investigated the corrosion behavior of samples prepared under different SPS process conditions in artificial simulated body fluids. Ag alloying can promote the formation of uniform, complete, dense, stable and thick passivation layer on the surface of Ti-Ta-Ag ternary alloy, which makes Ti-Ta-Ag ternary alloy uniformly corroded in Hank's solution without pitting. In addition, Ag alloying can effectively reduce the contact resistance of solid-liquid interface. These effects significantly enhance the corrosion resistance of Ti-Ta-Ag ternary alloy in Hank's solution. However, if there is a small amount of Ag precipitation on the surface of Ti-Ta-Ag ternary alloy, the trace precipitation of Ag just plays the opposite role to the above effect. The internal mechanism of the above phenomena is investigated by DFT calculation: on the Ag alloyed surface, Cl⁻ ions in Hank's solution are more strongly bound to the surface atoms, which is more conducive to the

formation and stability of dense passivation layer; while on the surface of Ag trace precipitation, Cl⁻ ions are directly bonded to the precipitated Ag atoms closely, forming isolated adsorption molecules similar to AgCl, which is adverse to the formation and stability of the passivation layer. Therefore, in order to improve the corrosion resistance of Ti-Ta-Ag ternary alloys, trace precipitation of Ag needs to be avoided as much as possible in the preparation.

Comment 10:

The presentation of this work should be further improved. Several sentences are not understood. A native English speaker is required.

Authors reply:

Thanks for this suggestion. We have carefully checked and revised the entire text to eliminate spelling and grammatical errors.

For Reviewer 2

Comments:

Dear authors, the attempt and the results are somewhat interests of ternary alloys for corrosion resistance application for engineering and automobiles. This may be okay; however, there are several data's need to be addressed such as surface morphologies of your materials. Anyhow, I accept this work at presently in this esteemed Journal..

Authors reply:

Thank you very much for your positive affirmation and support of our manuscript. This time, we have revised the manuscript, and hope it can continue to get your support and meet the publishing standards.

Detailed data on the preparation and characterization of Ti-Ta-Ag alloy samples (including process conditions, surface morphology, Ag trace precipitation, etc.) are the major subject of our published work [27. J.-M. Zhang, Z.-Y. Zhao, Q.-H. Chen, X.-H. Chen and Y.-H. Li, *RSC Adv.*, 2021, **11**, 2976-2984.]. And the main subject of this work is the corrosion resistance of these samples and their mechanism. Because of the different focus of the two parts of the work and the relatively large amount of data, we have divided it into two papers for publication.